# Mesenchymal Stem Cell Senescence and Osteogenesis

**DOI:** 10.3390/medicina58010061

**Published:** 2021-12-31

**Authors:** Artaria Tjempakasari, Heri Suroto, Djoko Santoso

**Affiliations:** 1Department of Internal Medicine, Faculty of Medicine, Airlangga University, Dr. Soetomo Teaching Hospital, Surabaya 60286, Indonesia; artnef@yahoo.com; 2Department of Orthopedic & Traumatology, Faculty of Medicine, Airlangga University, Dr. Soetomo Teaching Hospital, Surabaya 60286, Indonesia; heri-suroto@fk.unair.ac.id

**Keywords:** cellular senescence, mesenchymal stem cells, MSCs senescence, osteogenesis

## Abstract

Mesenchymal stem cells (MSCs) are stem cells with the potential ability to differentiate into various cells and the ability to self-renew and resemble fibroblasts. These cells can adhere to plastic to facilitate the culture process. MSCs can be used in research into tissue biotechnology and rejuvenation medicine. MSCs are also beneficial in recipient tissue and differentiate as a breakthrough strategy through paracrine activity. Many databases have shown MSC-based treatment can be beneficial in the reduction of osteogenesis induced by senescence. In this article, we will discuss the potential effect of MSCs in senescence cells related to osteogenesis.

## 1. Introduction

Senescence is a time-dependent functional decline that affects most organisms and is an important risk factor for human diseases such as malignancy, glucose metabolism disorder, cardiovascular disease, and neurodegenerative process [1]. Cellular senescence can be defined as persistent cell cycle termination associated with stereotyped phenotypic changes [2,3,4]. The cellular senescence process can be altered in response to external stimuli, including the lessening of telomere length, oxidative stress, deoxyribonucleic acid (DNA) injury, and oncogene activation [5].

Mesenchymal stem cells (MSCs) are known as multiple, mature, non-hematopoietic stem cells collected separately from bone marrow [6]. MSCs have been harvested and extracted from different tissues and organs, such as peripheral blood, umbilical cord, bone marrow, Wharton’s Jelly, placental tissue, breast milk, and other growth contributing organs, via different methods [7,8,9].

In order to achieve a minimal capacity to differentiate between osteocytes, adipocytes, and chondroblast in vitro, the International Society for Cellular Therapy (ISCT) has established minimum criteria for development of MSCs such as good adherence to plastic and fibro-blastoid, good immunophenotypic expression of cluster of differentiation (CD) 73, CD90, CD105, and minimal expression of CD34, CD45, CD14, CD19, CD79a, and human leukocyte antigen—DR isotype (HLA-DR) surface markers [10]. MSCs derived from these tissues show heterogeneity in biological features and functional abilities related to proliferative capacity, potentiation of multi-lineage derivative, proangiogenic ability, and immunomodulatory activity [11].

Senescence in tissue or organs is related to the loss of regenerative ability and functional deterioration, both of which are hypothetically associated with stem cells’ activity. Senescence may also be associated with the decline of stem cells, senescence of cells, premature aging, compromised renewal ability, or even skew differentiation [12].

Inside a normal bone, an osteogenesis process is regulated by biological stages involving MSCs, leading to the modeling and partial remodeling process, causing proliferation [13]. Other processes also occur, such as lineage differentiation, expression of specific markers, extracellular matrix (ECM) mineralization, and collagen expression. In aging bone, other known culprits than bone resorption activity are MSC impairment, shifting of osteogenesis to adipogenesis, and decreased capacity for renewal activity [14,15]. This imbalance activity may increase the risk of fractures [16]. Transcription factors in normal conditions are needed in MSC differentiation to maintain normal osteogenesis in a well-fashioned mechanism. The sequential activation of Runx2 and Osterix transcriptions is the master regulator of osteogenesis. At the same time, CCAAT enhancer-binding protein beta (CEBPβ), gamma (CEBPγ), alfa (CEBPα), and peroxisome proliferator-activated receptor-gamma (PPARγ) are the master regulators of adipogenesis [17,18]. In the aging process of the bone, there might be an imbalance related to these transcription factors.

As the senescence process continues, osteoblasts slowly decline in number, leading to a positive net bone resorption. Similar to humans, mice’s trabecular and cortical bone mass shows a decrease in cortical porosity and thickness due to the imbalanced remodeling process. This low bone formation is positively associated with a decrease of osteoprogenitor found inside the bone [19].

Decrease of Nicotinamide Adenine Dinucleotide (NAD+) in oxidized form when aging is a co-enzyme that functions as the electron acceptor in young tissues [20,21]. Nicotinamide mononucleotide (NMN) and nicotinamide riboside (NR) are NAD precursors which work as an anti-aging agent in tissues such as the pancreas and muscle vasculature [22,23]. Most of the cellular NAD+ is recovered from nicotinamide via the salvage pathway. Raising the NAD+ level is associated with low events of DNA damage, mitochondrial dysfunction, cell senescence, and stem cell degradation in many mice tissue studies via NAD-dependent Sirtuin 1 activity (Sirt1). Sirt1 deacetylates of FoxOs and β-catenin stimulate Wnt signaling and osteo-blastogenesis from osteoblast progenitors. An increase in reactive oxygen species (ROS) in the mesenchymal lineage, such as stem cells, progenitor of osteoblasts, and osteocytes, contributes to bone formation loss, thereby decreasing osteoblastogenesis [24]. This activity is stimulated by FoxOs activity and binding of the FoxOs to β-catenin. Therefore, Sirt1/FoxOs/β-catenin in osteoprogenitor activity contributes to skeletal aging as a significant inhibitor of bone formation [25,26,27].

Senescence features of MSCs show hyper-glandular and enlarged morphology, deficiency of proliferation and differentiation capacity, secretion of senescence-associated secretory phenotype (SASP), and changes of nuclear morphology and formation called senescence-associated heterochromatic foci (SAHF) [28,29,30]. MSC aging can also be measured with β-galactosidase activity, gene expression markers, length of telomere, gene methylation, and epigenetic markers [31]. MSCs experience a decline in functional capacity as well as an age increase. This will lead to decrease of tissue homeostasis, aging related disease, and even organ failure. In this article, we will discuss the potential effect of MSCs in senescence cells related to osteogenesis.

## 2. Cellular Senescence Process

### 2.1. Cell Cycle Arrest

The number of stimuli that cause aging gradually increases, and the mechanisms involved are extensively studied. These stimuli are signaled through various signaling pathways, many of which activate p53 (encoded with TP53 in humans and Trp53 in mice), all of which are essentially cyclin-dependent kinase (CDK) inhibitors. Agents are p16 (also known as INK4A; encoded by CDKN2A), p15 (also known as INK4B; encoded by CDKN2B), p21 (also known as WAF1; encoded by CDKN1A) and p27 (encoded by CDKN1B). Inhibition of the CDK-cyclin complex leads to growth arrest. A critical factor in implementing aging is the hypo-phosphorylated RB (retinoblastoma family) [32]. This accumulation leads to sustained RB family protein activity, inhibition of E2F transactivation, and cell cycle arrest. It is irreversible when proteins of the RB or p53 family are later inactivated [33]. These efforts are supported by the heterogeneity of E2F target genes [34], the action of cytokines secreted by senescent cells [35], and the increased production of long-lived ROS [36].

### 2.2. Decreased Telomere Length and Response to DNA Injury

Telomeres function like a molecular clock that records the replication history. In particular, “erosion” of telomeres due to sequential cell division that cannot preserve telomere length can lead to reduced telomere length and “replicative senescence” type. The loss of telomeres is recognized as DNA damage. It thus triggers a DNA damage response (DDR) similar to ionizing radiation and chemotherapy drugs. Telomeres are also highly susceptible to external DNA damage [37,38]. This is partly due to the inaccessibility of telomeres to DNA damage repair machinery, from yeast to humans [39]. Key mediators of DDR are the DNA damage kinases related to phosphorylation such as ATM, ATR, CHK1, and CHK2. The activation of several cell cycle proteins, including phosphorylation p53, activates the expression of p21, which binds and inhibits several CDK-cyclin complexes, particularly those involving CDK2 [5].

### 2.3. CDKN2A Locus Depression

Duplications are also associated with the CDKN2A locus (also known as INK4A and ARF), which encodes two important tumor suppressor factors, p16 and ARF. ARF regulates the stability of p53 by inactivating the p53-degrading protein MDM2 ubiquitin ligase E3 [40,41]. The CDKN2A locus is usually expressed at insufficient levels in new tissues but is repressed with age [42]. Although the molecular mechanisms responsible for inhibition of CDKN2A are not fully understood, it is well known that they are highly dependent on loss of the Polycomb inhibitor complex [43,44,45]. It should be clear that DNA damage can lead to degradation by reducing the level of ARF protein [46].

### 2.4. Stress-Inducing Senescence and ROS

ROS levels rise after various types of stress, including chemotherapy drugs, loss of telomere defenses, DNA damage, and oncogene activation [39,43]. A role for aging-related oxidative stress is evidenced by antioxidant treatment delaying or preventing aging [47,48,49]. Mechanistically high levels of intracellular ROS induced by the RAS-RAF-MEK-ERK cascade activate p38 MAPK to increase p53. transcriptional activation and p21 activation [40].

### 2.5. Aging Related Oncogene

Normal cells respond to the activation of many oncogenes by cellular senescence. Oncogene-induced by senescence was first detected in the oncogenic form of ASD in human fibroblasts. The list of oncogenes that can cause aging has grown to about 50. Aging caused by oncogenes occurs in vivo and is well known to act as a brake in the early stages of carcinogenesis. Inhibition of the CDKN2A locus is a common hallmark of oncogenic aging [41,42].

In addition, this type of senescence may also strongly induce DDR due to DNA damage caused by abnormal DNA replication [50,51] and ROS [5,39,40,41,42,43]. The relative importance of these mechanisms (p16, ARF, or p53 induced by DDR) is cell type dependent. In mice, the ARFp53 pathway is an important activator of oncogene-induced senescence [52], whereas in humans the DDRp53 pathway appears to play a more important role than the ARFp53 pathway [53]. Finally, p16 plays a small role in stimulating senescence in mice but is essential in human cells [54].

### 2.6. Senescence-Associated Secretory Phenotype (SASP)

Senescent cells produce a complex pro-inflammatory response known as the SASP and IL-8. Chemokines (monocyte chemoattractant protein (MCP) and macrophage inflammatory protein (MIP)), growth factors (transforming growth factor (TGF-β) and granulocyte-macrophage colony-stimulating factor (GMCSF)), proteases [55,56,57,58,59] and secretome aging messages (SMS) [60,61] are also included.

Secretion of these and similar proteins by senescent cells induces inflammation and, at least in some cases, may be necessary for phagocytotic senescent cell clearance [62,63]. SASP components, especially TGF-β, can also induce senescence of adjacent cells in a paracrine manner through mechanisms that generate ROS and DNA damage [64].

## 3. Mesenchymal Stem Cells (MSCs)

MSCs, also known as stromal cells, are a collection of tissue-specific progenitor cells that can renew in long-term and potential differentiation as an important role in tissues and organs balance [65,66,67,68,69,70]. These cells coexist and overlap due to plasticity differentiation and support of the functioning tissue. This depends on the source of the tissue, donor characteristics, culture media, and administration methods, “stem” or “stromal” [67]. MSCs, in this term, become a long reservoir for the next generation of somatic cells and other supernumerary cells.

MSCs can be isolated in large clusters from many sources such as tissues in bone marrow, perinatal, and adipose, and can be expanded by ex vivo means. The adhesion ability to plastic can be defined with a set of phenotype markers such as CD73+, CD90+, CD105+, CD11b- or CD14-, CD19- or CD79a-, CD34-, CD45-, and HLA-DR-. The definition is also not limited to the capacity of differentiation towards chondrocytes, adipocytes, and osteoblasts [71].

Senescence in organisms is correlated with a decrease of MSC activity which implies the declining of stem cell functions. This slowing activity reduces tissue repair and maintenance speed, a characteristic of senescence. As an example, fractures in osteoporotic bone associated with advanced age are more prone to delay in healing because of diminished function and amount of MSCs [72].

MSCs may be an optimizing option in the regenerative medicine aspect and tissue repair, with immunomodulatory benefit because of a convenient method of isolation and replication [73,74,75]. Paracrine activity in MSCs via soluble factor, exosome and micro-vesicles may also help ease tissue modulation in the microenvironment, inhibiting inflammation and contributing to the repair process [66,76,77].

MSCs secrete many soluble factors that work as autocrine or paracrine, including chemokines, proteases, extracellular matrix (ECM) growth factors, and cytokines, possibly used as cell-free-based therapy sources. Multiple functions such as pro-proliferative activity, anti-inflammation, pro-angiogenic, anti-apoptotic, and anti-fibrotic functions are due to the interaction between cells and secretion of abundant soluble factors. Anti-inflammatory secretome activity releases prostaglandin E2, Transforming growth factor- β (TGF-β), IL-6, IL-1, Tumor necrosis factor- inducible gene six protein (TSG6), IL-1 receptor antagonist (IL-1RA), and nitric oxide [78].

## 4. The Roles of Extracellular Vesicles

Other substances such as hepatocyte growth factor (HGF), fibroblast growth factor (FGF), and vascular endothelial growth factor (VEGF) are correlated with paracrine activity in MSCs. These molecules work in the extracellular compartment and are secreted via extracellular vehicles (EVs) for direct communication within the cells’ pathway. The EVs are separated into two subtypes: exosomes and micro-vesicles (or microparticles) [79]. Originating from the endosomal reserve compartment, exosomes (size less than 120 nm) produce multivesicular bodies (MVB), which fuse with the plasma membrane to secrete the content. Micro-vesicles (100–500 nm) are budding vesicles that come from the plasma membrane after stress induced by many conditions. Heterogeneity of EV size may result in the separation of exosomes from the vesicle, which becomes hard through EV isolation because of the physicochemical property existing between them. Large-size exosomes and small-size micro-vesicles have similar densities and dimensions, which is hard to purify because available purification methods can only separate small and large EVs irrespective of the biogenesis process [80]. EVs consist of nucleic acid (mRNA, DNA, microRNAs, and long non-coding RNAs), lipids, and proteins. The EVs are enhanced with specific lipids (such as sphingolipids and cholesterol) and proteins (such as tetraspanins and heat shock proteins).

EVs give rise to phospholipid-walled particles via encapsulation with cytosol and are important as cross-talk between cells in the physiologic and pathological condition in MSCs. EVs interact with ECM and the cells themselves when outside the extracellular medium. EVs also bring enzymes responsible for remodeling processes, such as metalloproteinases and regulators, or directly via secretion of the matrix-remodeling enzyme from the environment of the cells. Overall, this process may contribute to tetraspanins and proteoglycans, which alter the interaction between EVs and cells and modify ECM composition, which drives the interaction. The interaction influences the size of EVs and the acceptor cell’s physiological state [81].

The EVs’ capability to communicate can be recognized differently via activation of a specific receptor on the cell surface or direct delivery into the cell. EVs may secrete TLR9 on the surface of inactive macrophages, recognized by an activated macrophage. Another mechanism is release of chemokine TNF-α. Release of more EVs in MSCs may stimulate a different process inside the MSC itself [2].

## 5. Characteristics of Senescent MSCs

MSCs in senescence may have a poor colony efficiency rather than the earlier passage of MSCs [21]. MSC senescence is classically marked by the arrest of growth in the G1 phase of the cell cycle, expression of senescence-associated β-galactosidase (SA-β-Gal), change of morphology size, and senescence-associated lysosomal a-L-fucosidase (SA-a-Fuc) with alteration in surface marker [14].

Senescent MSCs face morphological changes, including more abundant actin fibers, reduction of bond capability to plastic surface, more flattened, hypertrophic and constricted nuclei, and more granular cytoplasm. To measure this, a colony forming unit-fibroblast (CFU-F) assay may be used and lipofuscin clumping may be increased after using autofluorescence. When MSCs are cultured inside dishes with low density plate culture, CFU-F may adhere and proliferate. The number of colonies may indicate ability to proliferate, while senescent MSCs decrease the amount of CFU-F. SA-β-Gal can be used for biomarker in aging. Detection using histochemical process was first described in 1995 and become a common assay used to determine the senescence of MSCs. Increase of SA-β-Gal is positively associated with increased aging via lysosomal activities and change in pH cytosolic [82,83]. The use of SA-a-Fuc has the potential ability to become a novel senescence marker which increases in response to DNA damage, replication, and oncogenic activity induced by senescence. It is even more sensitive and robust compared to SA-β-Gal in terms of transcriptional process and enzymatic levels detection [84].

Classically, MSC is defined by ISCT with these minimal criteria: (i) may adhere to plastic media; (ii) multiple differentiation potential of adipocytes, chondrocytes, and osteoblasts in standard culture conditions in vitro; (iii) expression of surface markers CD73, CD90, CD105 and absence of CD11b, CD14, CD15, CD19, CD34, CD79a, and HLA-DR [68]. However, these markers are equally expressed in young and senescent MSCs, increasing the challenge for differentiating senescent MSCs from the young [85]. Stro-1 [27], CD106 (VCAM-1) [86], and CD146 (MCAM) [66] expressions have potential uses as positive and negative markers of senescence during longstanding culture. CD106 especially is strongly downregulated in MSCs after chondrogenesis, osteogenesis, and adipogenesis. Therefore, this marker is beneficial in undifferentiated MSCs within MSC cultures. CD295 (leptin receptor) increases intrinsic cellular aging, suggesting a marker in apoptotic cells [87]. It can be concluded that MSCs can be divided into two groups. Stably expressed groups are CD73, CD90 and CD105, which may have little benefit regarding senescence status, while Stro-1 or CD106 rely on dependency donor, culture passages and other culture properties. The switching ability of MSCs into osteogenesis or adipogenesis via differentiation is mediated through signaling pathways, and transcription factors alter. One study shows that MSCs’ low activity in osteogenesis may increase the life span of organisms in line with the decrease in bone formation efficiency [88]. In bone-formation markers, alkaline phosphatase (ALP) and osteocalcin (OC) reduce the expression of senescence cells inside the culture medium [89].

DNA markers in senescent cells show nuclei containing heterochromatin with small and distinct spots called SAHF [90]. The spots represent dense chromatin, which is transcriptionally inactive, and downregulation of transcription factors such as cyclin A and E2F family member [90]. SAHF is identified via DAPI staining, heterochromatin finding associated histone markers, and high expression of H3K9me3 and H3K27me3 [91]. An increase in both of these inhibitory markers, as previously mentioned, may decrease gene expression.

Regulation of epigenetic alteration involves histone modification and cellular senescence, monitored via modification. It can be achieved via DNA methylation as the common process in MSCs’ senescence [92,93]. Association between hypomethylation and senescence occurs at the region of genome heterochromatic. This occurrence may interfere with transcription factors such as transposons, methylated CpG binding proteins, repetitive elements, and activation of the silenced gene [94]. There is also a decrease in gene-related senescence such as KDM3a-b, KDM5d, and KDM6a-b (part of lysine-specific demethylase) [95]. Therefore, senescence associate DNA methylation (SA-DNAm) can monitor cellular senescence in a specific gene and histone modification [91].

Through the paracrine effect, senescence cells potentiate the effect of neighboring cells, and this process is called the senescence-associated secretory phenotype (SASP) [96,97]. SASP factors consist of IL-1, IL-6, IL-8, matrix metalloproteinase 1 (MMP1), TNF-α, vascular endothelial growth factor (VEGF), and many others [98,99]. Micro-vesicles (MVs) are important for cell secretome components in senescence cells, which play important roles in regulating immunomodulation and suppressing tumor growth [35,100].

Telomere erosion limits the MSC division in the senescence process, becoming the hallmark of DNA damage in cells [101]. Response of DNA damage itself is associated with cell cycle arrest and senescence. TTAGGG repeat of chromosome telomerase can halt telomere erosion and promote elongation of the telomere. Telomerase reverse transcriptase (TERT) overexpression is related to increase in lifespan in animal experimental [102].

Meanwhile, EV secretion from senescent cells is partially dependent on the p53 pathway and the downstream target gene via tumor suppression-activated pathway 6 (TSAP6). P53 has an essential role as a gene-regulator of transcription, including Rab5B and Rab27B. These two regulators are vital in exosome biosynthesis and regulators of endosome [103]. EVs’ enhanced secretion is explained by two possible mechanisms. First, EVs mediate the cytoplasmic removal of DNA, which is not needed by the cell, misfolded, or recognized as a toxic molecule, enhancing cell survival. The fragmented DNA will activate DNA damage response (DDR) and is then exported by the EVs to prevent the deviation of the DDR activation pathway itself. As seen in MSCs, senescent cells release EVs as a defense mechanism to be marked as a distress signal, allowing nearby cells to respond to stress efficiently [104]. Secondly, EVs released from senescence cells are modulated via chronic systemic inflammation during the aging process, which somehow progresses in aging-related disease. miRNAs may be responsible (as so-called inflammamiRs) for aging-related processes, DDR, oxidative stress, mitochondrial dysfunction, and proteotoxic stress environment [105]. MSCs miRNA in turn is also modulated with the aging increase; therefore, decreased miRNA expression may correlate with the MSC-EVs’ aging process [106,107].

## 6. Osteogenesis in Healthy and Senescent MSCs

Aging in tissue and organ stages is related with stem cells. In human and animal research, aging impacts MSCs via a decreased series of MSCs within the bone marrow, and bias differentiation into adipocytes, which sacrifice osteoblasts. MSCs, or stromal mesenchymal cells, can grow in culture plate, proliferate in vitro, and differentiate into osteoblasts, chondrocytes and adipocyte. In addition, MSCs have been isolated from fats, pulp, amniotic fluid, placenta, and Wharton’s jelly [108].

Skeletal tissue MSCs are composed of bone and cartilage in response to growth factors such as bone morphogenetic proteins (BMPs) and Wnt molecules. MSCs express the bone morphogenetic transcription factors Runx2 and Osterix (Osx), which differentiate into osteoblasts. MSCs can express Sox9 and differentiate into cartilage, thereby forming chondrocytes. MSCs can also express CCAAT/enhancer-binding protein (C/EBPa) and peroxisome proliferator-activated receptor (PPAR-γ), which differentiate into bone marrow adipocytes [109,110].

In addition to osteoblasts, bone additionally incorporates osteoclasts that act as bone resorption elements from the HSC monocytic lineage. HSCs are pluripotent stem cells within the bone marrow that may produce all forms of blood cells. HSC-derived monocytes can grow to become macrophages and granulocytes, similar to osteoclasts. Osteoclasts are giant cells with many nuclei that secrete proteases which wreck bone matrix proteins and collagen. In addition, osteoclasts act synergistically with osteoblasts via complicated binding mechanisms. For example, MSC and osteoblasts secrete MCSF, RANKL, and OPG to modify osteoclast formation, and monocyte and osteoclasts secrete numerous boom elements to alter osteoblast formation. Osteoblast-mediated osteogenesis and osteoclast-mediated bone resorption are reservoirs of equilibrium. Osteogenesis outweighs bone resorption during a boom, and bone mass increases [111,112]. Senile osteoporosis is ordinarily because of a lower quantity of MSCs within the bone marrow and a lower osteogenesis, due to the differentiation of distorted MSCs into adipocytes at the price of osteoblasts [89]. MSC is known to have an osteogenesis and adipogenesis differentiation capability which is altered in older MSCs. Older MSCs tend to differentiate into adipocytes, thus the markers of osteogenesis, such as alkaline phosphatase (ALP) activity and osteocalcin (OC) expression, were down-regulated in aged MSCs during culture in the bone-forming media [113].

Aging can directly affect osteogenesis by preventing proliferation and inhibiting the function of MSCs, which can differentiate into a wide variety of cell populations, including osteoblasts and adipocytes [14,33,114]. Induction of aging is primarily regulated by the p53 and retinoblastoma pathways (pRb/p16INK4a) [115,116]. Expression of p16INK4a and the presence of lesions due to unresolved DNA damage are the best markers of aging in vivo [42,117,118]. Both pathways are closely associated with bone homeostasis. For example, pRb is directly involved in the differentiation of bone progenitor cells because it can bind to and activate major osteogenesis regulators such as Runx2 [119,120]. pRb can also suppress adipogenic differentiation through a mechanism of action on the peroxisome proliferator-activated receptor-γ subunit (PPAR-γ) [121]. Decreased p53 or p21CIP1 regulators may increase the likelihood of mouse stromal cell proliferation and osteogenesis differentiation [118,121,122]. Increased expression of the osteogenesis transcription factors Runx2 and Osterix (Osx) may be the underlying mechanism of control of osteogenesis by p53.

Several studies have shown that the osteogenesis activity of MSCs deteriorates with increasing lifespan, which is associated with decreased osteogenesis efficiency. This osteogenesis is associated with the expression of RUNX2/CBFA1 via the PI3KAKT signaling pathway. It is an important transcription factor of the osteogenesis/chondrogenic lineage as an activator and marker of MSC osteogenesis [123,124]. A slight decrease in its expression has been observed with age [125]. The core factor-kB ligand-receptor activator (RANKL), which is essential for osteoclast differentiation and maintenance, has been highly expressed in late MSC. Transforming growth factor (TGF/SMAD3) signaling pathways are essential for osteogenesis differentiation and may induce ERK phosphorylation. ERK inhibitors have been shown to suppress TGFβ-induced osteogenesis differentiation [126,127].

Potentiation of adipogenic MSCs’ capacity is relative and may be worsened or enhanced. The general view is that the adipogenesis activity of MSCs may decrease in conjunction with usual culture media. PPARµ is a member of the ligand-activated nuclear receptor superfamily, and may act as adipogenic-specific transcription factor, including transcriptional activation. This PPARµ targets different genes related to lipid metabolism and adipocyte development. The expression may be decrease, alongside with senescence, and the impairment of this expression along with C/EBP may alter the fate of MSCs’ osteoblast lineage. The inhibition pathway of C/EBP and PPARµ is mediated via WNT/β-catenin signaling and is therefore aimed at MSC differentiation into osteoblasts. Hence, it can become the key regulator of adipogenesis and osteogenesis [54]. Phosphorylation of AKT by insulin may suppress the expression of Forkhead box O3 (FOXO3) and activate PPAR, which opposes the balance of differentiation and enhances adipogenesis activity.

FOXOs play an essential role in bone turnover and osteoclast activation by decreasing the ROS substance [59]. FOXO1, 3, and 4 deletions in the osteoclast progenitor increase osteoclast proliferation and formation, thus reducing the trabecular and cortical bone mass. On the contrary, the increased function of FOXO3 in turn inhibits the osteoclast differentiation and increases the survival of osteoblast via catalase and superoxide dismutase. The last two enzymes mentioned prevent oxidative injury occurrence [47,60]. FOXO1 also helps collect glutathione, which decreases ROS via the sulfhydryl moieties redox-active pathway [128].

The PI3K-PKB/AKT regulates the transcriptional activity of the FOXOs via the canonical pathway. FOXOs and Sirt1 are related to the increase in bone lifespan through the balance of bone formation and resorption, while IGF1 and IGF-R1 act oppositely. Sirt1 modifies posttranslational FOXOs and prevents bone turnover while enhancing bone formation. Wnt signaling and insulin-like signaling (ILS) are reduced with FOXOs activity. The decreased signaling of Wnt may induce protein aggregation, which contributes to early cell deterioration [129].

The Insulin/Insulin Growth Factor 1 (IGF-1) signaling system (IIS) regulates metabolism, including activity concerning an organism’s nutrition balance, growth, and development. Mammals have three sepae ligand molecules of insulin /IGF-1 receptors: insulin, IGF-1, and IGF-2. There are also three diverse insulin/IGF tyrosine kinase receptors: insulin receptor (IR), IGF-1 receptor (IGF-1R), and the so-called orphan IR related receptor (IRR). The activated IGF-1 or insulin receptors begin with phosphorylation in intracellular substrates, then dock for intracellular effectors. The site of this docking process for intercellular effectors includes growth-factor-receptor-bound protein-2 (Grb2) and the p85 regulatory subunit of PI-3K. Activation of Ras-MAPK and PI-3K-PKB/AKT pathway occurs as 2 major signals. The last pathway mentioned has been known to regulate insulin/IGF-1 signaling metabolic effect [130].

Sirtuin1 (Sirt1)3 is an NAD+-dependent deacetylase that delays and opposes aging-related processes in lower organisms in mammals [131,132]. Sirt1 is responsible for biological activity such as DNA repair, metabolism of energy, mitochondrial homeostasis, and tumor suppression. This activity is linked with the deacetylation of the FOXO family and β-catenin, which acts as co-activator of canonical Wnt signaling [133].

Osteoblasts are terminally differentiated and have a short life span, which explains the need for continuous replacement from a mesenchymal progenitor to maintain bone growth and development [134]. This replacement activity is maintained by proliferation and differentiation of progenitor osteoblast, expressing Runx2 and Osterix1 (Osx1). Then, the progression of this committed progenitor into mature osteoblast is regulated by Wnt signaling [135]. Canonical Wnt signaling is first modulated by binding Wnt ligands to Frizzled and low-density lipoprotein receptor related to 6 cell membrane receptors. This activity halts the destruction of β-catenin and promotes its accumulation. Inside the nucleus, β-catenin is associated with the T-cell factor (TCF)/lymphoid-enhancing factor (Lef) family of transcription factors and enhances the expression of contributing genes involved in proliferation and differentiation. Wnt signaling can be downregulated with an extracellular and intracellular inhibitor. Then, the FOXOs further downregulate the Wnt signaling via β-catenin binding and prevent the β-catenin with TCF/Lef [136]. The alleviation of β-catenin/TGF transcription via FOXOs in osteoblast progenitors halts bone development, which decreases the mass itself [137].

Reduction of NAD+ somewhat in line with senescence of osteoblast progenitors was associated with an increase in CD38 expression (main nicotinamide nucleotidase found in mammalian tissues) [137]. CD38 is a protein with a multifunction process involved in the formation of the second messengers ADPR and cyclic-ADPR (cADPR), related to signaling-dependent-calcium intracellular formation. Along with NADase activity, CD38 also contributes significantly to the homeostasis of NAD inside cellular and tissue. This finding is supported by inhibition, via pharmacology or genetics, of CD38 increased NAD+ in multiple mice organs. Therefore, the senescence-related NAD+ process becomes slower, which results in attenuation in mitochondrial dysfunction and improvement of glucose tolerance [138,139].

The level of Nicotinamide phosphoribosyl-transferase (Nampt) protein inside osteoblast cells from old mice was lower than in young mice. Along with this finding, deletion of Nampt in mesenchymal lineage cells is associated with a significant decrease in bone density and supports the hypothesis that NAD+ plays a role in osteoblast progenitor but may retard bone formation. This finding is strengthened via NR administration, which may increase the osteoprogenitor number and mineralization of bone in aging mice [140].

NR administration decreases the acetylation of FOXOs and β-catenin related age. Further decrease of NAD levels via pharmacological or genetic means correlates with further enhancement of acetylation of both proteins. Acetylation of FOXOs will increase β-catenin and decrease Wnt signaling, therefore decreasing osteo-blastogenesis [137,141]. Otherwise, Sirt1 deacetylates FOXOs and β-catenin, which provide the activation of osteo-blastogenesis [142,143]. In mice, osteoblast cells lacking FOXO1, 3, and 4 are protected partially from FK866, supporting the evidence that Srt1/FOXOs may mediate the effect of NAD+ in the osteo-blastogenesis process. The administration of Srt1 agonist in mice diminished skeletal aging [144].

## 7. Conclusions

Cellular senescence displays a decreased capacity of MSCs and permanent cell cycle arrests. Osteogenesis differentiation decreases over time, whereas adipogenesis increases over time. Decrease in osteoblast progenitors and osteoblasts itself contribute to loss of bone formation. The role of each molecular mechanism of MSCs in aging remains unclear. Intrinsic factors including signaling pathways and effector molecules, and extrinsic factors including systemic factors and niche molecules, may control the senescence process of MSCs. Further studies are needed to unravel the exact functions of these factors. MSC networks and delays in MSC aging may inhibit senescence development at the cellular level.

## Data Availability

The study did not report any novel data.

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
