# Peer review of "Mesenchymal Stem Cell Senescence and Osteogenesis"

_medicina, 2021, doi:10.3390/medicina58010061_

Round 1

Reviewer 1 Report

In this review, Tjempakasari and co-authors recalled many information about cell senescence with an interest on mesenchymal stem cell (MSC) senescence. Indeed, MSCs are of clinical interest, given their role for example in immunomodulation and regeneration,  thus MSC senescence is surely a relevant issue.  

However I have some concerns. Although the bibliography on cell senescence is quite recent and updated, I was expecting a review on mesenchymal stem cells and their modification during senesce, as claimed by the title and the abstract. Surprisingly, the review is a comprehensive dissertation on cell senescence in general, and only chapter 5 (of 10!) concerns about modifications mesenchymal stem cells. Overall the review sounds redundant and many concepts are repeated. Actually, even sometimes also sentences are repeated (see lines 238-239 and 341-342 for example). At the end the reader miss the point of the review. Since the topic is of interest I suggest the authors to synthetized the generalized concepts on cell senescence to focus on MSC changes. How does senescence impact on MSC communication with other cells? Does senescence change immunomodulatory capacity of MSC? What about relationship with cancer? Does the extracellular vesicle production change? There are really many aspect that could be deepen. Otherwise it is a general review on cell senescence, and in that case the title and the abstract must be changed.

Minor: Is “squandering of proteostasis “ a common name for alteration of proteostasis?

Author Response

In this review, Tjempakasari and co-authors recalled many information about cell senescence with an interest on mesenchymal stem cell (MSC) senescence. Indeed, MSCs are of clinical interest, given their role for example in immunomodulation and regeneration, thus MSC senescence is surely a relevant issue. 

However, I have some concerns. Although the bibliography on cell senescence is quite recent and updated, I was expecting a review on mesenchymal stem cells and their modification during senesce, as claimed by the title and the abstract. Surprisingly, the review is a comprehensive dissertation on cell senescence in general, and only chapter 5 (of 10!) concerns about modifications mesenchymal stem cells. Overall the review sounds redundant and many concepts are repeated. Actually, even sometimes also sentences are repeated (see lines 238-239 and 341-342 for example). At the end the reader misses the point of the review. Since the topic is of interest I suggest the authors to synthetized the generalized concepts on cell senescence to focus on MSC changes. How does senescence impact on MSC communication with other cells? Does senescence change immunomodulatory capacity of MSC? What about relationship with cancer? Does the extracellular vesicle production change? There are really many aspects that could be deepen. Otherwise it is a general review on cell senescence, and in that case the title and the abstract must be changed.

We thank the Reviewer for the critical assessment of our manuscript. Following the Reviewer’s suggestions and comments, we have now restructured the entire manuscript, removing the redundant parts of the manuscript and added the suggested contents about MSC changes and communications. We also added a discussion about the immunomodulatory capacity of MSC. Finally, we also discussed briefly about extracellular vesicle production change as suggested.

“The EVs are separated into two subtypes: exosomes and microvesicles (or microparti-cles) (80). Originating from the endosomal reserve compartment, exosomes (size less than 120 nm) produce multivesicular bodies (MVB), which fuse with the plasma membrane to secrete the content. Microvesicles (100-500 nm) are budding vesicles that come from the plasma membrane after stress induced by many conditions. Heteroge-neity of EV size may result in the separation of exosomes from the vesicle, which be-comes hard through EV isolation because of the physicochemical property between them. Large-size exosomes and small-size microvesicles have similar densities and sizes, which is hard to purify because available purification methods can only separate small and large EVs irrespective of the biogenesis process (81). EVs consist of nucleic acid (mRNA, DNA, microRNAs, and long non-coding RNAs), lipid, and proteins. The EVs are enchanted with specific lipids (such as sphingolipids and cholesterol) and proteins (such as tetraspanins and heat shock proteins).

EVs, give rise to phospholipid-walled particles via encapsulation with cytosol and are important as cross-talk between cells in the physiologic and pathological condition in MSCs. EVs interact with ECM and the cells themselves when outside the extracellular medium. EVs also bring enzymes responsible for remodeling processes such as met-alloproteinases and regulators, or directly via secretion of the matrix-remodeling enzyme from the environment of cells. Overall, this process may contribute to tetraspanins and proteoglycans, which alter the interaction between EVs and cells and modify ECM composition, which drives the interaction. The interaction influences the size of EVs and acceptor cell physiological state (82).

The EVs capability to communicate can be recognized differently via activation of specific receptor on cell surface or direct delivery into the cell. EVs may secrete TLR9 on surface of inactive macrophages to be recognized by activated macrophage. An-other mechanism is release of chemokine TNF-α. Release more EVs in MSCs may stimulate different process inside the MSCs itself (83).” Pages 4 and 5 Lines 193-219.

Minor: Is “squandering of proteostasis “ a common name for alteration of proteostasis?

We apologize for this error. We have removed it from our manuscript.

Reviewer 2 Report

The authors review the main features, processes and detection methods  of senescence focusing on mesenchymal stem cells. Although the topic proposed should be interesting the review is not well organized and the title only partially reflects the content of the review. 

Paragraphs 4 and 5 on mesenchymal stem cells should include more detailed and precise information relatively to the specific mechanisms through which senescence occurs and affects stemness and differentiation.

Paragraph 6 and 7 are too short and not informative

Paragraph 8.2 does not consider lysosomal changes

In general the information are approximate, the review is redundant and the english language requires an extensive editing

Author Response

The authors review the main features, processes and detection methods of senescence focusing on mesenchymal stem cells. Although the topic proposed should be interesting the review is not well organized and the title only partially reflects the content of the review.

Paragraphs 4 and 5 on mesenchymal stem cells should include more detailed and precise information relatively to the specific mechanisms through which senescence occurs and affects stemness and differentiation.

Paragraph 6 and 7 are too short and not informative

Paragraph 8.2 does not consider lysosomal changes

In general the information are approximate, the review is redundant and the english language requires an extensive editing

We thank the Reviewer for the critical assessment of our manuscript. In response to those comments and taking into account the comments from other Reviewers, we have now restructured our manuscript, putting more emphasis on the MSCs and osteogenesis, rather than the global review on the cellular senescence. We hope that our efforts to improve the quality of our manuscript suffice.

Reviewer 3 Report

A large number of studies were reported in the past years that senescent cells were involved in lots of cellular processes or disorders. It is meaningful to summarize how senescent MSCs impair or contribute to osteogenesis. However, the authors need to describe more detail and more focused on MSCs and osteogenesis. In current version, the authors spent too much space to describe the regular biomarkers of aging. Therefore, the authors need to focus on the newest studies between MSCs and osteogenesis. Meanwhile, as we all know now that NAD declined with aging and was involved in lots of age-related diseases, including osteogenesis. Thus, NAD metabolism should also be summarized in the following version.

Author Response

A large number of studies were reported in the past years that senescent cells were involved in lots of cellular processes or disorders. It is meaningful to summarize how senescent MSCs impair or contribute to osteogenesis. However, the authors need to describe more detail and more focused on MSCs and osteogenesis. In current version, the authors spent too much space to describe the regular biomarkers of aging. Therefore, the authors need to focus on the newest studies between MSCs and osteogenesis. Meanwhile, as we all know now that NAD declined with aging and was involved in lots of age-related diseases, including osteogenesis. Thus, NAD metabolism should also be summarized in the following version.

We thank the Reviewer for the important suggestions and remarks. Taking into account the comments from all Reviewers, we have now restructured the manuscript, putting more emphasis on the MSCs and osteogenesis. We also, among others, accommodated the excellent suggestion to discuss NAD metabolism in the manuscript.  

“Decrease of Nicotinamide Adenine Dinucleotide (NAD+) in oxidized form when aging is a co-enzyme that functions as the electron acceptor in young tissues (20,21). Nico-tinamide mononucleotide (NMN) and nicotinamide riboside (NR) are NAD precursors which work as an anti-aging agent in tissues like the pancreas and muscle vasculature (22,23). Most of the cellular NAD+ is recovered from nicotinamide (NAM) via the salvage pathway. Raising the NAD+ level is associated with low events of DNA damage, mitochondrial dysfunction, cell senescence, and stem cell degradation in many mice tissues studies via NAD-dependent Sirtuin 1 activity (Sirt1).” Page 2 Lines 62-69.

“Reduction of NAD+ somewhat in line with senescence of osteoblast progenitors was associated with an increase in CD38 expression (main nicotinamide nucleotidase found in mammalian tissues) (151). CD38 is a protein with a multifunction process involved with the formation of the second messengers ADPR and cyclic-ADPR (cADPR) related signaling-dependent-calcium intracellular formation. Along with NADase activity, CD38 also majorly contributes to the homeostasis of NAD inside cellular and tissue. This finding is supported by inhibition via pharmacology or genetic of CD38 increase NAD+ in multiple mice organs. Therefore, the senescence-related NAD+ process becomes slower, which results attenuation in mitochondrial dysfunction and improvement of glucose tolerance (152,153).” Page 9 Lines 420-429.

Round 2

Reviewer 2 Report

The authors have improved the review and have more focused on mesenchymal stem cells as suggested. However, the parts on MSCs and osteogenesis are too descriptive and not very related to the MSCs senescence mechanisms and consequences.

Authors should consider to move EVs in a new paragraph as well as to better organize the entire review. In addition, many abbreviations are spelled out several times, there are some typos like senescence-associated-heterochromatic-foci (line 79) and many part should be rephrased with particular attention on English language and style.

Author Response

The authors have improved the review and have more focused on mesenchymal stem cells as suggested. However, the parts on MSCs and osteogenesis are too descriptive and not very related to the MSCs senescence mechanisms and consequences.

We thank the Reviewer for the comments and suggestions, which have helped us to improve the quality of our manuscript. We have now rechecked our manuscript as per Reviewer's suggestion.

Authors should consider to move EVs in a new paragraph as well as to better organize the entire review. In addition, many abbreviations are spelled out several times, there are some typos like senescence-associated-heterochromatic-foci (line 79) and many part should be rephrased with particular attention on English language and style.

As suggested, we have moved the EV part to a separate section (Section 4) and we also performed another check on the typo and errors. Regarding the mentioned typo in line 79, it seems that it was due to the conversion from the original Word document to MDPI PDF format. We could not find it in the DOCX version of the revised manuscript.

Once again, we really appreciate the overall comments provided by the Reviewer and we hope that we could now accommodate most of those concerns satisfactorily. 

Reviewer 3 Report

I have no more comments.

Author Response

Once again, thank you for your invaluable comments. They have helped improving our manuscript significantly.